# High-Throughput RNA Sequencing Reveals *NDUFC2*-AS lncRNA Promotes Adipogenic Differentiation in Chinese Buffalo (*Bubalus bubalis* L.)

**DOI:** 10.3390/genes10090689

**Published:** 2019-09-06

**Authors:** Jieping Huang, Qiuzhi Zheng, Shuzhe Wang, Xuefeng Wei, Fen Li, Yun Ma

**Affiliations:** 1College of Life Sciences, Xinyang Normal University, Xinyang, Henan 464000, China; 2School of Agriculture, Ningxia University, Yinchuan, Ningxia 750021, China

**Keywords:** *Bubalus bubalis* L., adipose, RNA sequencing, long noncoding RNAs, adipogenesis

## Abstract

The buffalo (*Bubalus bubalis* L.) is prevalent in China and the increasing demand for meat production has changed its role from being a beast of burden to a meat source. The low fat deposition level has become one of the main barriers for its use in meat production. It is urgent to reveal factors involved in fat deposition in buffalo. This study performed RNA sequencing to investigate both long noncoding RNAs (lncRNAs) and mRNAs of adipose tissues in young and adult buffalos. A total of 124 lncRNAs and 2008 mRNAs showed differential expression patterns between young and adult samples. Coexpression analysis and functional enrichment revealed 585 mRNA–lncRNA pairs with potential function in fat deposition. After validation by qRT-PCR, we focused on a lncRNA transcribed from the ubiquinone oxidoreductase subunit C2 (NDUFC2) antisense (AS) strand which showed high correlation with thyroid hormone responsive protein (THRSP). *NDUFC2*-AS lncRNA is highly expressed in adipose tissue and maturation adipocytes and mainly exists in the nucleus. Functional assays demonstrated that *NDUFC2*-AS lncRNA promotes adipogenic differentiation by upregulating the expression levels of *THRSP* and CCAAT enhancer binding protein alpha (C/EBPα) in buffalo. These results indicate that *NDUFC2*-AS lncRNA promotes fat deposition in buffalo.

## 1. Introduction

Adipogenesis is influenced by a multitude of factors. In addition to a number of protein-coding genes [1], noncoding RNAs, such as microRNAs (miRNAs) [2] and long noncoding RNAs (lncRNAs) [3], can be prominent modulators of adipogenesis. Protein-coding genes and miRNAs are well studied and they are highly conserved across species. lncRNAs are a class of transcripts with more than 200 nucleotides that are not translated into proteins. They can act as gene expression regulators with relatively low conservation [4,5]. Though multiple lncRNAs have been identified as significant regulatory factors of adipogenesis, most of them were studied in rodents and humans. The first lncRNA (steroid receptor RNA activator) was reported to enhance adipogenesis and adipocyte function through multiple pathways in both ST2 mesenchymal precursor cells and 3T3-L1 cells [6]. Several other lncRNAs had been discovered to regulate adipogenesis, such as lncRNA NEAT1 in 3T3-L1 cells [7] and Blnc1 in mice [8]. In humans, lncRNA ADINR promotes adipogenesis by activating the transcription of CCAAT enhancer binding protein alpha (C/EBPα) [9]. The lncRNA H19 inhibits adipocyte differentiation during the commitment of bone marrow mesenchymal stem cells into adipocytes [10]. The lncRNA MEG3 was reported to be involved in the balance between adipogenic and osteogenic differentiation in human adipose-derived stem cells [11]. Recently, several lncRNAs that modulate adipogenesis in pigs and cattle have been identified. Sirtuin 1 (Sirt1) AS lncRNA was found to promote *Sirt1* translation via combination with Sirt1 mRNA, forming an RNA duplex in pigs [12]. ADNCR was shown to suppress adipogenic differentiation by targeting miR-204 in cattle [13]. To date, lncRNAs involved in adipogenesis and even the characteristics of lncRNAs expression profile have not been reported in buffalos.

Buffalos (*Bubalus bubalis*) are abundant in China [14]. Traditionally, buffalos were raised for draught power. In recent years, the utility of buffalos in agriculture has gradually decreased due to increasing agricultural mechanization, which suggests there can be a conversion of the role of buffalos into a meat source. Fat deposition is one of the most important traits for meat animals. However, fat deposition level in buffalo is very low due to the long-term breeding for draught power. Thus, investigation of lncRNAs involved in fat deposition is important for buffalo breeding.

In this study, high throughput RNA sequencing of adipose tissues was performed using the Illumina HiSeq 3000 platform. Tissues were obtained from buffalos at different development stages. Coexpression analysis and functional enrichment were performed to yield candidate lncRNAs with a putative role in fat deposition. Further qRT-PCR validation and functional assays demonstrated that a lncRNA, which transcribed from the ubiquinone oxidoreductase subunit C2 (NDUFC2) antisense (AS) strand, promotes adipogenic differentiation in buffalo adipocytes. This study provides transcriptome information for further studies on buffalo fat deposition and proposes novel biomarker for aiding in the improvement of fat deposition in buffalo breeding.

## 2. Materials and Methods 

### 2.1. Animal Ethics

All animal protocols were approved by the Institutional Animal Care and Use Committee (IACUC) of Xinyang Normal University. All animals were raised according to the feeding and management standards of buffalo production, and all efforts were made to minimize suffering.

### 2.2. Animals and Tissue Samples

Xinyang buffalos (bull, *n* = 6) were produced by different female animals and share the same male parent. They were born within a month of each other and were randomly selected and equally divided into two groups (young group and adult group). They were raised at the Xinyang buffalo farm (Xinyang, Henan, China) under similar feeding and management conditions. Animals were weaned at 6 months of age and the animals in the young group (*n* = 3) were then slaughtered. The remaining three individuals in the adult group received a diet of 3 kg/day concentrate until 12 months of age, 4 kg/day concentrate until 24 months of age, and followed by 4.5 kg/day concentrate until 30 months of age. Forage was provided ad libitum. Individuals in the adult group were slaughtered at 30 months of age (*n* = 3). Subcutaneous adipose tissue was sampled after slaughter and immediately frozen in liquid nitrogen for RNA sequencing and qRT-PCR validation. In addition, other 50 buffalos with variable months of age were also sampled for qRT-PCR validation assay.

For primary adipocyte isolation, fresh adipose tissue was sampled, kept at ~30 °C in phosphate buffer saline (PBS) with 1% streptomycin and penicillin, and taken back to lab for isolation and culture of adipocytes.

### 2.3. RNA Isolation and Sequencing

For RNA sequencing, two groups were designated, young (6-month-old, *n* = 3) and adult (30-month-old, *n* = 3). Total RNA was extracted using TRIzol (Invitrogen, Carlsbad, CA, USA), following the manufacturer’s instructions. RNA quantity was measured with NanoDrop2000 (Nanodrop, Wilmington, DE, USA) and 1.5% agarose gels. RNA with 1.8 < 260/280 value < 2.0 and concentration > 500 ng/μL was used for further analysis. rRNA was removed from the total RNA using the Epicentre Ribo-zero rRNA Removal Kit (Epicentre, Madison, WI, USA). The rest of the RNA was fragmented and reverse transcribed by adding random primers to yield double stranded cDNA. Subsequently, end repair, poly (A) tailing, adapters ligating, and PCR enrichment (with 10 cycles) experiments were successively performed to generate the cDNA library. The cDNA was purified with ethanol and the DNA quality was assessed using the Agilent Bioanalyzer 2100 system (Agilent Technologies, Palo Alto, Cali, USA). Clustering of the sample was performed using the Quant-iT™ PicoGreen® dsDNA Assay Kit (Life, Grand Island, NY, USA) according to the manufacture’s instruction. Finally, the cDNA library was sequenced using the paired-end sequencing mode of the Illumina HiSeq 3000 platform (Illumina, San Diego, CA, USA). The RNA sequencing data were deposited in the Genome Expression Omnibus of NCBI. The accession number is GSE112744.

### 2.4. Quality Control and Transcriptome Assembly

Low-quality reads (>50% of the bases had Phred quality scores ≤ 10) and those containing adapters were removed to obtain clean reads using Trim Galore (Version 0.4.2, Babraham Institute, Combridge, UK) [15]. The Phred scores, including Q20 and Q30, length of reads, and GC contents of each read were calculated. High-quality data were used for the subsequent analysis.

Since annotation information for buffalo genome (UOA_WB_1) was not available, the cattle genome (UMD3.1) was used. The cattle reference genome and gene model annotation files were downloaded from the Ensemble database (http://www.ensembl.org/index.html). Each read was aligned to the reference genome using STAR [16]. Mapped reads were assembled using Cufflinks v2.2.1 [17]. Cufflinks was run with the following settings: ‘min-frags-per-transfrag = 0’ and ‘-library-type fr-firststrand’. All other parameters were set to default.

### 2.5. Coding Potential Analysis

The mRNAs were transcripts that contained in the known genes. For lncRNAs prediction, transcripts that were shorter than 200 bp, those with multiple exons, and those with an ORF of more than 100 amino acids were removed. All transcripts were classified by Cuffcompare [17], based on its position in the reference genome. The following four types of transcripts were considered as primary lncRNAs: intergenic transcripts, transcripts overlapping known introns, antisense transcripts overlapping known exons, and antisense transcripts overlapping known introns. Three tools were used to assess the coding potential of the remaining transcripts, including CPC [18], PhyloCSF [19], and CPAT [20]. Transcripts with a CPC score < 0, a PhyloCSF score < 0, and a CPAT score ≤ 0.364 were retained. Transcripts with a potential Pfam protein domain were filtered using HMMER [21].

### 2.6. Differential Expression Analysis and Functional Enrichment

Fragments per kilobase of transcript per million mapped reads (FPKM) was used as an index to calculate the expression level of each transcript in every library and was calculated using Cuffdiff 2.1.1 [17]. The expression level was indicated as log2FPKM. When the absolute value of log2(fold change) was ≥1 and the FDR value ≤ 0.05, the transcript was considered as differentially expressed (DE) lncRNA or mRNA in the two groups and was presented in heatmap prepared with the R package [22].

DE mRNAs were used for functional enrichment analysis. DAVID (Version 6.8, https://david.ncifcrf.gov/) was used for the gene ontology (GO) analysis with a hypergeometric test to investigate the molecular function. KOBAS (Version 3.0, http://kobas.cbi.pku.edu.cn/index.php) was used for KEGG enrichment analysis with a hypergeometric test to identify the pathways [23]. A *p* value ≤ 0.05 was used as threshold to evaluate significant enrichment.

### 2.7. Coexpression Analysis

To explore the potential target genes of the DE lncRNAs, coexpression analysis was performed between DE mRNAs and lncRNAs. mRNA–lncRNA pairs with Pearson’s correlation coefficients |*r*| > 0.95 were retained. mRNAs involved in the pairs were considered as potential target genes of lncRNAs.

### 2.8. 5′- and 3′-Rapid Amplification of cDNA Ends (RACE)

To identify the full-length sequence of *NDUFC2*-AS lncRNA, RACE experiments were performed by the SMARTer RACE cDNA Amplification Kit (Clontech, Palo Alto, CA, USA) following the manufacturer’s protocol. Total RNA was isolated from adipose tissue in buffalo. The gene-specific primers (GSP) used for 5′ and 3′ RACE were 5′-CTCCCGCCTCCAGCCCAGAACCT-3′ and 5′-CTCAACCCAGCTTCCCAACCAGGGA-3′, respectively.

### 2.9. Adenovirus Packaging

The full-length of *NDUFC2*-AS lncRNA was obtained by using overlap PCR with the production of 5′ and 3′ RACE (Appendix A) and was ligated to pMD18-T vector (TaKaRa, Dalian, China). Recombinant pMD18-T vector was sent to Hanbio Biotechnology Co.; Ltd. (Shanghai, China) for overexpression adenovirus packaging. Briefly, adenoviral vectors were created using the AdMax system, including the backbone plasmid pHBGloxΔE1, 3cre, and the shuttle plasmid pHBAd-EF1α-MCS-CMV-EGFP. EGFP was used as indicator for transduction efficiency. Full-length *NDUFC2*-AS lncRNA was contained in Ad-*NDUFC2*-AS lncRNA. Ad-GFP was used as negative control.

### 2.10. Isolation and Culture of Buffalo Primary Adipocytes

Buffalo primary adipocytes were isolated using the tissue block method [24]. Briefly, about 3 mm^3^ sections of adipose tissue without visible blood vessels and fascia were cut and cultured in a 10 cm cell culture dish. The cell culture dish was inverted in an incubator set to 37 °C with 5% CO_2_ for 6 h without medium. Then, 8 mL of high glucose Dulbecco’s Modified Eagle Medium (Hyclone, Logan, UT, USA) containing 20% fetal bovine serum (Hyclone, Logan, UT, USA) and 1% streptomycin and penicillin (Hyclone, Logan, UT, USA) was added. The cells were incubated at 37 °C with 5% CO_2_ for about 15 days. The adipose tissues were then removed and primary adipocytes were digested and collected for further culture.

### 2.11. Adenoviral Transduction

Buffalo primary adipocytes were planted in six-well plates in triplicate. Transduction was conducted when the primary buffalo adipocytes reached 80% confluence. Adenovirus Ad-*NDUFC2*-AS lncRNA and Ad-GFP were added to cells at the indicated multiplicity of infection (MOI), respectively. The media was exchanged 3 h later.

### 2.12. Adipogenic Differentiation, Oil Red O Staining, and Quantification

Two days after adenovirus transduction, primary adipocytes were treated with inducing medium containing 10 μg/mL insulin (Sigma, Milwaukee, WI, USA), 1 μΜ dexamethasone (Sigma, USA), 0.5 mM IBMX (Sigma, Milwaukee, WI, USA), and 1 μΜ rosiglitazone (Sigma, Milwaukee, WI, USA) for 2 days. Cultures were then treated with a maintenance medium containing 10 μg/mL insulin and 1 μΜ rosiglitazone. The medium was changed every two days.

After inducing with adipogenic agents for 6 days, Oil Red O staining was performed. Primary adipocytes were washed with PBS thrice and treated with two changes of 10% formalin for 5 min and 1 h, respectively. The cells were then washed with 60% isopropanol and stained with 0.3% Oil Red O (0.3% Oil Red O, 60% isopropanol, and 40% PBS) for 20 min. Finally, cells were washed with PBS five times and observed under the microscope.

For quantification of lipid accumulation in the cells, 100% isopropanol was used to elute the Oil Red O. Then, the spectrophotometric absorbance of Oil Red O was quantified at 510 nm with 100% isopropanol used as blank solution.

### 2.13. qRT-PCR

Primers were designed using the Pick Primers function from NCBI (http://www.ncbi.nlm.nih.gov/tools/primer-blast/) (Appendix A). Glyceraldehyde-3-phosphate dehydrogenase (GAPDH) was used as the internal control gene, the primers of which were reported in a previous study [13]. The total RNA was transcribed into cDNA using the PrimeScriptsRT reagent Kit with gDNA Eraser (TaKaRa, Dalian, China). qPCR was performed using SYBR Green I (TaKaRa, Dalian, China) with two-step reactions according to the manufacturer’s recommended protocol. The cycle threshold (2^−ΔΔCt^) method was used to calculate the relative expression level of lncRNA. Three replicates were run per sample and the qRT-PCR experiment was performed three times.

### 2.14. Statistical Analysis

Normal distribution testing was performed for the data. Comparison was analyzed by SPSS software (version 19.0, IBM, Armonk, NY, USA). Student’s *t* test was used when the data had a normal distribution, otherwise nonparametric test was used. A *p* value < 0.05 was considered to indicate statistical significance. Results are presented as mean ± SD (*n* = 3) by Origin software (version 7.5, Origin Lab, Wellesley, MA, USA).

Expressional correlation analysis between lncRNA and mRNA was performed using the CORREL function by Excel software. Meanwhile, correlation between lncRNA and mRNA is presented by scatter plot, showing formula and r value.

## 3. Results

### 3.1. Overview of RNA Sequencing

Six cDNA libraries of adipose tissues were constructed and sequenced. In total, 133,216,720 to 224,667,746 raw reads and 130,254,066 to 217,117,252 clean reads were obtained (Appendix A). A total of 141.50 Gb, with an average of 23.58 Gb clean data, was obtained (Appendix A). All clean reads were aligned to the cattle genome (UMD3.1) and the mapped ratios ranged from 76.00% to 86.05% (Appendix A).

### 3.2. Differentially Expressed Transcripts

The expression levels of all transcripts were used for the correlation analysis of both samples to evaluate the reliability of our data. High correlation was detected (*r* > 0.86), indicating the high reproducibility of the utilized method (Figure 1a). The sequencing data were used for mRNA and lncRNA analyses.

A total of 21,693 mRNAs were obtained, 2008 of which were identified as DE mRNAs in both groups (1021 up- and 987 downregulated) (Appendix A). Well-known adipogenesis markers, such as the peroxisome proliferator-activated receptor gamma (PPARG) [25], lipoprotein lipase (LPL) [26], and thyroid hormone responsive protein (THRSP) [27] were consistently upregulated in the adult group, which is to be expected given the animals development of adipose tissue at this age. For lncRNAs prediction, a total of 9494 lncRNAs were obtained (Appendix A), including 6512 intergenic lncRNAs (69%), 625 intronic lncRNAs (7%), 584 antisense lncRNAs (6%), 114 bidirectional lncRNAs (1%), and 1659 unclassified lncRNAs (17%) (Figure 1b). Among these, 124 lncRNAs showed differential expression in both groups, with 52 down- and 72 upregulated lncRNAs (Appendix A). Hierarchical clustering of the DE mRNAs and lncRNAs could be used to accurately distinguish the young from the adult buffalos (Figure 1c,d). Together these results indicate that the RNA sequencing data obtained was reliable.

### 3.3. Validation of Differentially Expressed lncRNAs and mRNAs by qRT-PCR

To evaluate the reliability of the DE analysis results, ten DE mRNAs (upregulated) associated with lipid metabolism and 14 DE lncRNAs (six up- and eight downregulated) were randomly selected for validation by qRT-PCR. The expression patterns of the ten mRNAs and 14 lncRNAs were consistent with those obtained from RNA sequencing (Figure 2). These results suggest a high quality of the DE analysis results. Thus, DE mRNAs and DE lncRNAs could be used for the subsequent analysis.

### 3.4. Coexpression Analysis and Screening mRNA–lncRNA pairs Associated with Fat Deposition

To investigate the potential functions of DE lncRNAs, coexpression analyses for DE mRNAs and lncRNAs were performed based on Pearson’s correlation coefficients. Up to 5315 mRNA–lncRNA pairs with |*r*| > 0.95 were obtained (Appendix A). Among these, 2195 mRNA–lncRNA pairs showed negative correlation, while the remainder showed positive correlation.

Then, to obtain a gene list associated with fat deposition, functional enrichment analysis for DE genes (mRNAs) was performed. A total of 1000 gene ontology (GO) items and 38 Kyoto encyclopedia of genes and genomes (KEGG) pathways with significant values (*p* < 0.05) were identified (Appendix A). Among these, items associated with fatty acid metabolism, energy metabolism, lipid metabolism, and PPAR signaling pathways received particular focus. As a result, 34 GO items (Appendix A, from GO-1 to GO-34) and eight KEGG pathways (Appendix A, from KEGG-1 to KEGG-8) were further screened. In these items, 213 genes were involved, and we classified these as making up the putative fat deposition-associated gene set.

This gene set was then used to screen for mRNA–lncRNA pairs with potential effects on fat deposition. At last, a total of 585 mRNA–lncRNA pairs were retained, including 74 lncRNAs and 147 mRNAs (Appendix A). Notably, most of the retained pairs (381/585) had positive correlations. Several lncRNAs showed relatively high correlations with well-known adipogenesis markers, such as *PPARG*, *LPL*, and *THRSP*. Several adipogenesis markers shared the same lncRNAs and vice versa (Appendix A).

### 3.5. Validation of the Expressional Correlation between lncRNA and mRNA by qRT-PCR

To evaluate the reliability of the coexpression analysis, five mRNA–lncRNA pairs with potential effect on fat deposition were selected for validation via qRT-PCR. These included three pairs for the *THRSP* gene, one pair for the peroxisome proliferator-activated receptor alpha (PPARA) gene, and one pair for the LPL gene (Table 1). Firstly, six adipose tissue samples of young (*n* = 3) and adult (*n* = 3) buffalos were used. Only three pairs with high correlation (*r* > 0.8) were then used for the second validation experiment by using 50 adipose samples of buffalos with variable months of age. Finally, only two pairs with relatively high correlations obtained: the TCONS_00539210-*THRSP* had *r* = 0.78 and the TCONS_00539092-*THRSP* had *r* = 0.81 (Table 1 and Figure 3).

### 3.6. Characterization of the NDUFC2-AS lncRNA

We further focused on the mRNA–lncRNA pair with the highest correlation value, TCONS_00539092-*THRSP*. TCONS_00539092 is a lncRNA transcribed from the *NDUFC2* antisense strand and we named it *NDUFC2*-AS lncRNA (Figure 4a). Meanwhile, *NDUFC2*-AS lncRNA is located 3′-downstream of *THRSP* in genome (Figure 4a). The full-length of *NDUFC2*-AS lncRNA is 2493 bp (Appendix A). The CPC [18] indicated that *NDUFC2*-AS lncRNA is a noncoding RNA (Figure 4b). The semiquantitative PCR of nuclear and cytoplasmic fractions showed that *NDUFC2*-AS lncRNA is mainly localized in the nucleus of adipose tissue (Figure 4c). Tissue expression profile demonstrated that *NDUFC2*-AS lncRNA is mainly expressed in adipose tissue (Figure 4d). During adipocyte differentiation, *NDUFC2*-AS lncRNA is significantly upregulated in day 10 (Figure 4e, *p* < 0.01). *NDUFC2*-AS lncRNA and *THRSP* showed relatively high expressional correlation during differentiation of buffalo adipocytes (Figure 4f).

### 3.7. Effects of NDUFC2-AS lncRNA on Lipid Accumulation in Buffalo Adipocytes

To detect the effect of *NDUFC2*-AS lncRNA on lipid accumulation in buffalo adipocytes, the full-length of *NDUFC2*-AS lncRNA was inserted into the pHBAd vector and packaged into adenovirus for overexpression (Ad-*NDUFC2*-AS lncRNA). The GFP was used as indicator for the recombinant adenovirus. GFP was highly expressed two days after adenoviral transduction (Figure 5a). The expression of *NDUFC2*-AS lncRNA in the Ad-*NDUFC2*-AS lncRNA group was significantly higher than that in the Ad-GFP group (Figure 5d). With overexpressed *NDUFC2*-AS lncRNA, *THRSP* and *C/EBPα* were significantly upregulated (Figure 5f,h). The expression of *PPARG* was significantly upregulated on day 4 but significantly downregulated on days 2 and 6 (Figure 5g). For *NDUFC2*, no significant difference was found between the two groups (Figure 5e). Compared to the Ad-GFP group, lipid accumulation was significantly increased in Ad-*NDUFC2*-AS lncRNA group (Figure 5b,c) in the sixth day of induced differentiation (*p* < 0.05).

## 4. Discussion

This is the first report that presents a characterization of lncRNAs based on RNA sequencing data during the development of adipose tissue in the buffalo. This study demonstrates that: (1) a large number of lncRNAs are expressed during the development of buffalo adipose tissue, most of which are intergenic lncRNAs; (2) 124 lncRNAs demonstrate differential expression during both development stages; (3) 74 lncRNAs show high expressional correlation with genes involved in fat deposition; and (4) *NDUFC2*-AS lncRNA promotes lipid accumulation in buffalo adipocytes. The central idea and results of this research are illustrated in Figure 6.

### 4.1. Prediction and Differentially Expression of lncRNAs

The identification of lncRNAs in animals and other organisms has recently become a research hotspot. Structurally, lncRNAs show weak or no protein-coding potential and low exon numbers [28,29]. Based on these principles, a total of 9494 putative lncRNAs were obtained in this study (Appendix A), which is comparable to the results of other studies [13,30,31]. Matching similar studies [13,29], most of the identified lncRNAs were of the intergenic type in this study (Figure 1b).

To identify lncRNAs with potential function in regulating adipogenesis, characterization of lncRNAs profiles have been investigated in animals [32,33,34]. Differential expression was demonstrated for 1336 lncRNAs during preadipocyte differentiation in chicken [32]. A study investigated lncRNAs in adipose tissue and identified only 18 DE lncRNAs between castrated and intact male pigs [33]. By comparing the transcriptome of different types of adipocytes, 175 lncRNAs were identified to be specifically regulated during adipogenesis, of which a portion is bound by PPARG and C/EBPα at the promoter regions [34]. Downregulating ten of the lncRNAs resulted in a reduction in lipid accumulation and in the expression of key adipogenesis genes [34]. In this study, 124 lncRNAs were identified that were differentially expressed in adipose tissues of young and adult buffalos (Appendix A). These lncRNAs may play important roles by affecting the expression of genes that regulate the development of adipose tissue.

### 4.2. lncRNAs Associated with Fat Deposition

lncRNAs have a low level of cross-species conservation and lack distinct common sequence features or structural motifs, which impedes the prediction of their putative function and functional mechanisms [4,5]. A number of studies have indicated that lncRNAs act via genomic targeting, where they serve as functional *cis*- or *trans*-regulatory elements [35]. The prediction of target genes of lncRNAs by *cis* and *trans* algorithms has been widely used in similar studies [36,37,38,39,40]. The *cis* algorithm assumes that lncRNAs target neighboring genes (<100 Kb). Based on this algorithm, a considerable number of putative target genes were obtained in this study. However, further functional enrichment identified no item that is associated with fat deposition (data not shown). The *trans* algorithm suggests that lncRNAs and their target genes have high correlation in their expression levels. By combining the *trans* algorithm and functional enrichment, a total of 585 mRNA–lncRNA pairs were obtained which were considered to have potential effects on fat deposition in buffalos (Appendix A). Among these, several genes with key roles in fat deposition were included, such as PPARA and LPL [1]. These results indicate that the candidate lncRNAs may regulate the expressions of genes that with key roles in fat deposition. In addition, multiple genes were targeted by one lncRNA and vice versa (Appendix A), suggesting that fat deposition is regulated by a complex mRNA–lncRNA network.

To evaluate the quality of the mRNA–lncRNA pair set with a potential effect on fat deposition, five pairs were selected for validation by qRT-PCR. Only two pairs (TCONS_00539210-*THRSP* and TCONS_00539092-*THRSP*) showed a relatively high correlation (Table 1 and Figure 3). These results indicate that the accuracy of correlation is relatively low in the present study, which may be ascribed to the small sample size used for RNA sequencing. We further focused on the mRNA–lncRNA pair with the highest r value, *THRSP* and TCONS_00539092. TCONS_00539092, namely *NDUFC2*-AS lncRNA, overlaps two exons of *NDUFC2* (Figure 4a), suggesting that it may influence the splicing of *NDUFC2* and prevent the normal expression of NDUFC2 [41]. NDUFC2 is the first enzyme complex of the mitochondrial electron transport chain and is associated with some neurological disorders [42]. Until now, no research has indicated association between NDUFC2 and lipid metabolism. Besides, *NDUFC2*-AS lncRNA is located downstream of *THRSP*. THRSP has been studied in rats for its potential function in lipid metabolism [43,44]. Recently, *THRSP* was found to be expressed in mature adipocytes in cattle [45] where it showed positive and high correlation with intramuscular fat [46], suggesting it has a putative role as a transcriptional regulator in adipogenesis. *NDUFC2*-AS lncRNA is mainly located in the nuclei of adipose tissue (Figure 4c). When investigating the expression profile, *NDUFC2*-AS lncRNA was found to be highly expressed in adipose tissue and mature adipocytes (Figure 4d,e). All these signs indicated that *NDUFC2*-AS lncRNA might affect fat deposition by regulating the transcription of *THRSP* [47]. Expectedly, *NDUFC2*-AS lncRNA significantly upregulated *THRSP* (Figure 5f) and promoted lipid accumulation in buffalo adipocytes (Figure 5b,c). However, *NDUFC2*-AS lncRNA had no significant effect on the expression of *NDUFC2* (Figure 5e). During induced differentiation of adipocytes, we found that the expressional pattern of *NDUFC2*-AS lncRNA in normal cultured cells (Figure 4e) was different from that in Ad-GFP or Ad-*NDUFC2*-AS lncRNA cells (Figure 5d). Primary adipocytes are very sensitive to culture condition. Treating with adenovirus made cells grow poorly and promoted cell apoptosis. Thus, the highest expression level was found in day 10 of differentiation in normal cultured cells (Figure 4e) while that was found in day 2 of differentiation in Ad-GFP or Ad-*NDUFC2*-AS lncRNA cells (Figure 5d).

When investigating the expression of adipogenic markers, *NDUFC2*-AS lncRNA was found to significantly upregulate *C/EBPα* (Figure 5h), while exerting an ambiguous effect on *PPARG* (Figure 5g). PPARG and C/EBPα are the key transcription factors and play central roles in the complex regulatory network of adipogenesis [3]. They cooperate in promoter regions, regulating a wide range of genes that are expressed in the adipogenic differentiation of adipocytes [48,49]. Meanwhile, expressions of *PPARG* and *C/EBPα* can be regulated by an array of factors that are involved the adipogenic differentiation of adipocytes [3].

Therefore, *NDUFC2*-AS lncRNA promotes lipid accumulation by increasing the expression of *THRSP* and *C/EBPα*. It was reported that Flank10kb class lncRNA can interact with target gene via a *cis*-regulatory process [50,51]. *NDUFC2*-AS lncRNA significantly upregulates the expression of *THRSP* and is downstream of *THRSP* in the genome. Besides, *NDUFC2*-AS lncRNA was mainly detected in the nucleus. Thus, we suspected that *NDUFC2*-AS lncRNA might regulate *THRSP* through a transcriptional modification. It might bind to the promoter of *THRSP* directly or with the help of other protein involved in transcription. An RNA immunoprecipitation assay can be used to detect the direct interaction between *NDUFC2*-AS lncRNA and the promoter of *THRSP*. Then, the upregulated THRSP might further upregulate the expression of C/EBPα. However, to the best of our knowledge, though THRSP demonstrates significant function in lipid metabolism [43,44], no evidence indicates a direct relationship between C/EBPα and THRSP. Thus, further work is necessary to investigate the specific regulatory mechanisms of *NDUFC2*-AS lncRNA during lipid accumulation.

## 5. Conclusions

This study firstly presents the characterization of lncRNA expression profiles in the adipose tissue of both young and adult buffalos. The results suggest that *NDUFC2*-AS lncRNA promotes lipid accumulation by upregulating the expression of *C/EBPα* and *THRSP* in buffalo adipocytes, but the underlying specific regulatory mechanism requires further research. This study provides valuable information for further studies on buffalo fat deposition, which will aid buffalo breeding.

## Figures and Tables

**Figure 1 genes-10-00689-f001:**
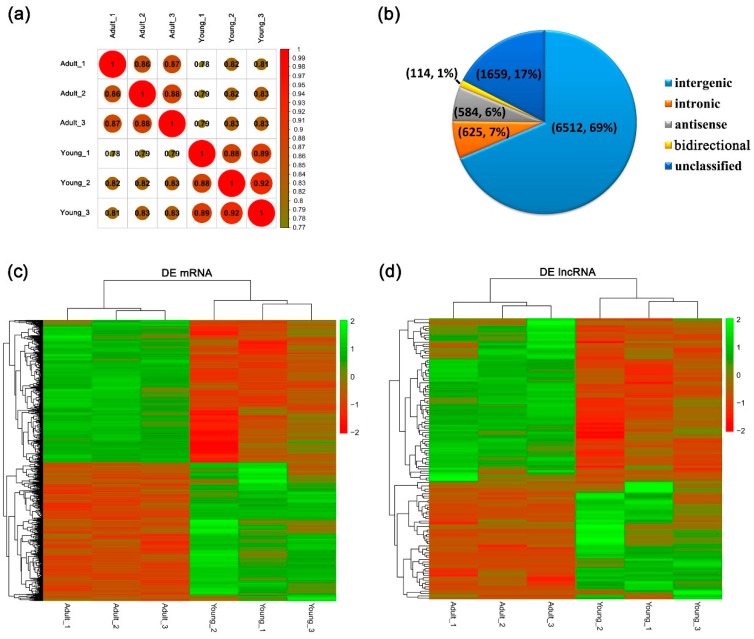
Global mRNA and lncRNA profiling of adipose tissues of young and adult buffalos. (**a**) Cluster graph of all mRNAs and lncRNAs based on correlation analysis. The correlation of mRNA and lncRNA expressions between both groups indicated the reliability of RNA sequencing performance. (**b**) Classification of the 9494 lncRNAs detected in this study. (**c**) Hierarchical clustering of 2008 differentially expressed (DE) mRNAs from young and adult groups. (**d**) Hierarchical clustering of 124 DE lncRNAs from young and adult groups. Data are presented as fragments per kilobase of transcript per million mapped reads (FPKM). Red: upregulated expression; Green: downregulated expression.

**Figure 2 genes-10-00689-f002:**
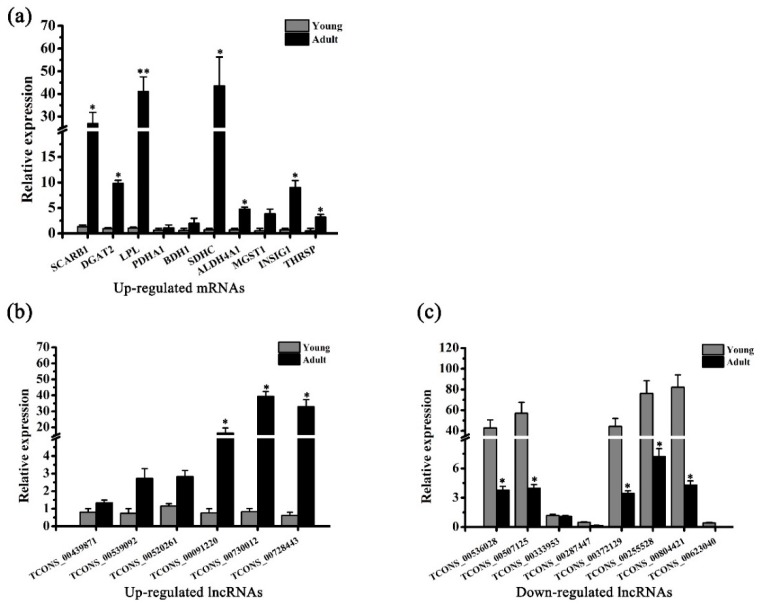
Validation of differentially expressed mRNAs and lncRNAs by qRT-PCR. (**a**–**c**) Expression patterns of ten upregulated mRNAs, six upregulated lncRNAs, and eight downregulated lncRNAs in adipose tissues of young and adult buffalos. The RNA expression levels are normalized to those of *GAPDH*. Comparison was analyzed by Student’s *t* test or nonparametric test. Data are presented as mean ± SD (*n* = 3, *, *p* < 0.05, **, *p* < 0.01).

**Figure 3 genes-10-00689-f003:**
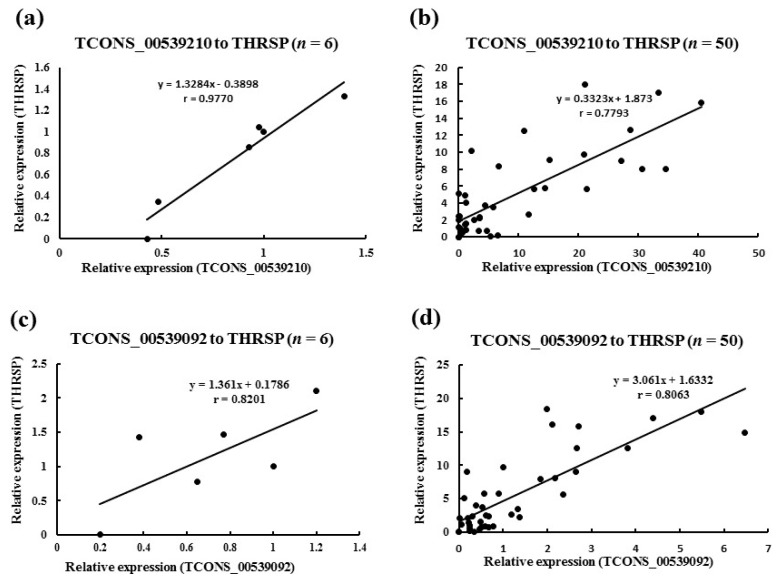
Validation of the two lncRNAs with high correlation to *THRSP* by qRT-PCR. The axis shows the relative expression of lncRNA and *THRSP*. Pearson’s correlation coefficients between lncRNA and *THRSP* based on their relative expressions were calculated to yield an r value. (**a**,**c**) Validation for TCONS_00539210-*THRSP* and TCONS_00539092-*THRSP* in young and adult buffalos, respectively (*n* = 6). (**b**,**d**) Validation for TCONS_00539210-*THRSP* and TCONS_00539092-*THRSP* in buffalos with variable months of age, respectively (*n* = 50).

**Figure 4 genes-10-00689-f004:**
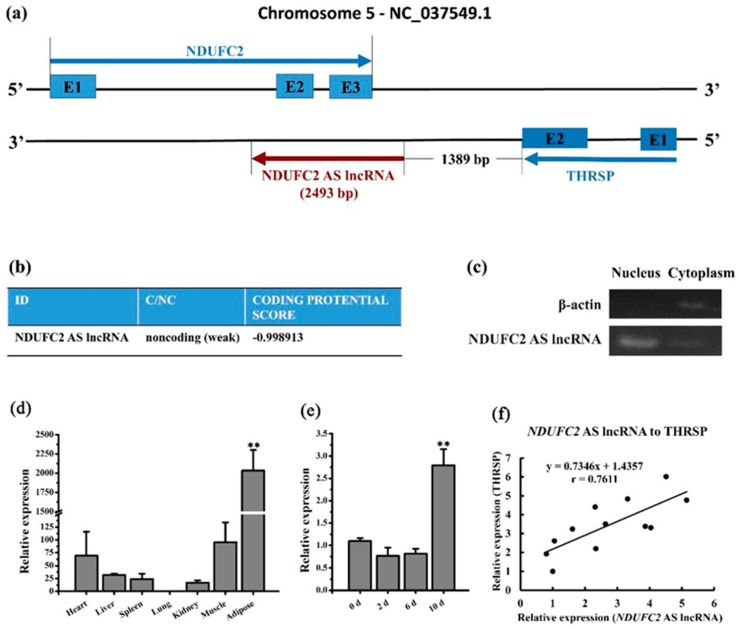
Characterization of *NDUFC2*-AS lncRNA. (**a**) Positional relationship among NDUFC2, *NDUFC2*-AS lncRNA, and *THRSP* in buffalo genome. (**b**) Protein coding ability prediction of *NDUFC2*-AS lncRNA by Coding Potential Calculator (CPC) program. (**c**) Cell localization of *NDUFC2*-AS lncRNA by semiquantitative PCR. RNA was isolated from nuclear and cytoplasmic fractions of adipose tissue. β-actin mRNA was used as control. (**d**) Tissue expression profile *NDUFC2*-AS lncRNA determined by qRT-PCR. (**e**) The expression dynamics of *NDUFC2*-AS lncRNA during adipocyte differentiation determined by qRT-PCR. Comparison was analyzed by Student’s *t* test or nonparametric test. Data are presented as mean ± SD (*n* = 3, *, *p* < 0.05, **, *p* < 0.01). (**f**) Expressional correlation analysis of *NDUFC2*-AS lncRNA and THARSP during differentiation of buffalo adipocytes. Adipocytes in day 0, day 2, day 4, day 6, and day 10 of induced differentiation were collected for qRT-PCR. There were three replicates for each stage.

**Figure 5 genes-10-00689-f005:**
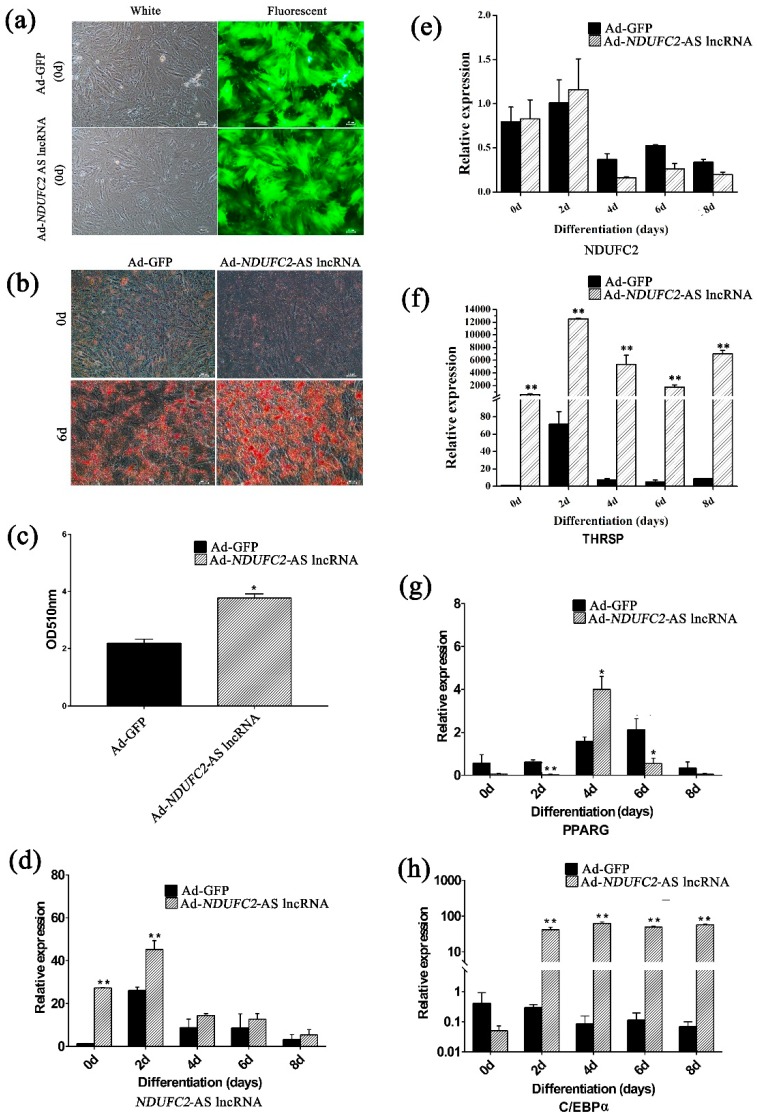
Overexpression of *NDUFC2*-AS lncRNA promotes lipid accumulation in buffalo primary adipocytes. (**a**) Micrographs of GFP-positive bovine adipocytes under white and fluorescent light in the Ad-GFP (control) and *NDUFC2*-AS lncRNA groups. Adipocytes were induced to differentiation 2 days after adenovirus transduction. 0d represents the time of differentiation. Scale bar, 100 μm. (**b**) Images of Oil Red O staining in buffalo adipocytes transfected with Ad-GFP and Ad- *NDUFC2*-AS lncRNA at day 0 and day 6 of adipogenic differentiation. Scale bar, 100 μm. (**c**) Histogram showing the quantitation of Oil Red O staining by spectrophotometry (normalized to control group). (**d**–**h**) The RNA expression dynamics of *NDUFC2*-AS lncRNA, *NDUFC2*, *THRSP*, *PPARG*, and *C/EBPα* during adipogenic differentiation in buffalo adipocytes transfected with Ad-GFP and Ad-*NDUFC2*-AS lncRNA. Comparison was analyzed by Student’s *t* test or nonparametric test. Data are presented as mean ± SD (*n* = 3, *, *p* < 0.05, **, *p* < 0.01).

**Figure 6 genes-10-00689-f006:**
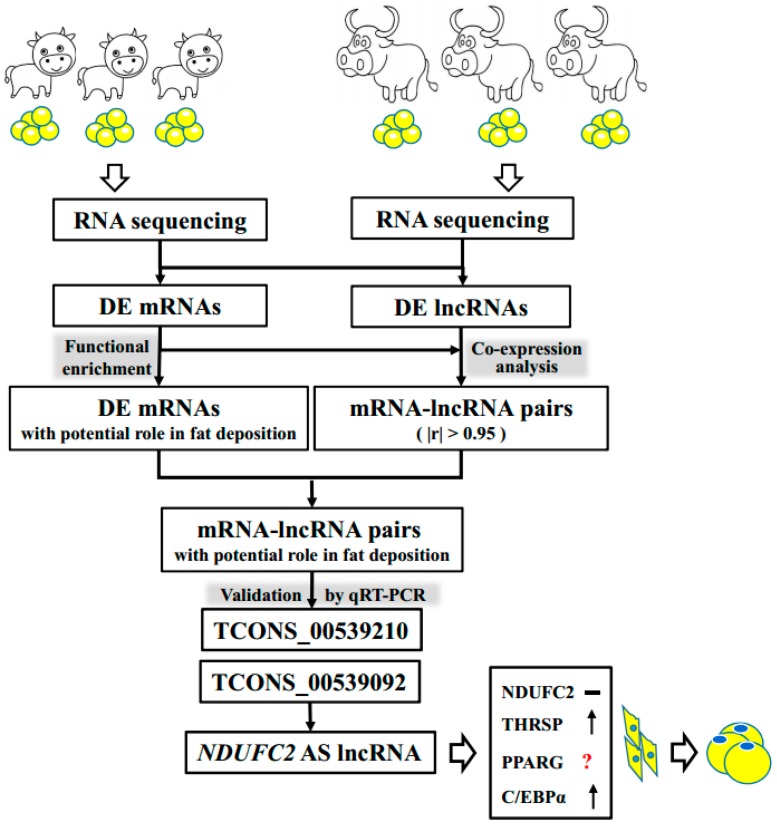
Schematic illustration of the experimental procedure and the main results.

**Table 1 genes-10-00689-t001:** mRNA–lncRNA pairs validation using qRT-PCR.

mRNA	lncRNA	*r* by RNA Sequencing	*r* by qRT-PCR	
		Young vs. Adult (*n* = 6)	Young vs. Adult (*n* = 6)	Adipose Samples of Different Months (*n* = 50)
*THRSP*	TCONS_00539210	0.99	0.98	0.78
*THRSP*	TCONS_00539092	0.98	0.82	0.81
*THRSP*	TCONS_00439871	0.96	0.51	NA
*PPARA*	TCONS_00539092	0.98	−0.06	NA
*LPL*	TCONS_00539092	0.97	0.94	0.14

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
