# Peer review of "High-Throughput RNA Sequencing Reveals *NDUFC2*-AS lncRNA Promotes Adipogenic Differentiation in Chinese Buffalo (*Bubalus bubalis* L.)"

_genes, 2019, doi:10.3390/genes10090689_

Round 1

Reviewer 1 Report

The presented manuscript: „High-throughput RNA sequencing reveals NDUFC2 AS lncRNA promotes adipogenic differentiation in Chinese Buffalo (Bubalus bubalis)” shows an interesting finding of the possible mechanisms related to adipocyte tissue development. Authors used the novel NGS sequencing method which allows monitoring all transcriptomic changes during adipocyte tissue development.  On the other hand, several details should be corrected and clearly explained.

Material and methods section - you should add the full description for reagents (company, city, country), for apparats (company, city, country - e.g. line 127), and for software (company, version).

Lines 69-76: According to a low number of animals, the detail information about the relationship between individuals should be shown. What was the gender of animals? Were they represent only one sex? If not the sex effect should be checked and included in statistical analysis.

Line 71 – ‘similar forage’ – what does it mean? Were animals feed with the same condition or not?

Line 87 – You should show the RIN values obtained for RNA samples and show the range of RIN which was acceptable for you as a sample taken for further analysis.

Lines 93-95: The NGS procedure should be explained in more details -  the number of cycles, type of NGS sequencing approach (single or pair-end seq.) and technical replicates, etc.

Line 121-122:  Please checked did you really use log2FC>2. Log2FC above 2 means that FC vales taken to analysis were above 4!  If you used such restrictive criteria please explain why.

Lines 186-188: The information of the statistical approach used to data analysis is very limited. Did you check the normality of distribution of obtained data?  You used t-student test analysis which can indicate that data has a normal distribution. On the other hand, you should use a non-parametric test.  The correlation used to mRNA-lncRNA validation using qPCR should be described.

Line 192: Did you verify why the differences in a number of reads per samples are so huge? Table S3 clearly show that young animals have average higher numer of raw reads compared to adult animals (195413343 vs 147820378). I know that you normalization algorithm, but as mapping ration shows that groups differ also in % of mapping 77.8% for young and 85.2% for adults. Please explain.

Author Response

Response to Reviewer 1 Comments

The presented manuscript: “High-throughput RNA sequencing reveals NDUFC2-AS lncRNA promotes adipogenic differentiation in Chinese Buffalo (Bubalus bubalis)” shows an interesting finding of the possible mechanisms related to adipocyte tissue development. Authors used the novel NGS sequencing method which allows monitoring all transcriptomic changes during adipocyte tissue development.  On the other hand, several details should be corrected and clearly explained.

Point 1: Material and methods section - you should add the full description for reagents (company, city, country), for apparats (company, city, country - e.g. line 127), and for software (company, version).

Response 1: Thank you for your professional comment. We have added the full description for (the main) reagents, for apparatus, and for software to the “Material and methods” section.

Point 2: Lines 69-76: According to a low number of animals, the detail information about the relationship between individuals should be shown. What was the gender of animals? Were they represent only one sex? If not the sex effect should be checked and included in statistical analysis.

Response 2: Thank you for your professional comment. Xinyang buffalos (bull, n = 6) were produced by different female buffalos but share the same father. They all born within a month of each other and were randomly selected and equally divided into two groups (young and adult groups). Relative information has been added to the “Animals and tissue samples” section.

Point 3: Line 71 – ‘similar forage’ – what does it mean? Were animals feed with the same condition or not?

Response 3: Thank you for your careful review. Animals were feed with the same condition. The sentence has been modified in line 73.

Point 4: Line 87 – You should show the RIN values obtained for RNA samples and show the range of RIN which was acceptable for you as a sample taken for further analysis.

Response 4: Thank you for your professional suggestion. RNA with 1.8 < 260/280 value < 2.0 and concentration > 500 ng/μL was used for further analysis. This sentence has been added in lines 88-89.

Point 5: Lines 93-95: The NGS procedure should be explained in more details -  the number of cycles, type of NGS sequencing approach (single or pair-end seq.) and technical replicates, etc.

Response 5: Thank you for your careful review and professional suggestion. Ten cycles were used for PCR enrichment in cDNA library construction (line 92). Paired-end sequencing mode was performed (line 97) in this study. According to the construction of manufacture, isothermal amplification was used for clustering (line 96). No technical replicate for the sequencing. Three biological replicates, namely three young and three adult buffaloes were used for the sequencing (lines 85-86).

Point 6: Line 121-122:  Please checked did you really use log2FC>2. Log2FC above 2 means that FC vales taken to analysis were above 4!  If you used such restrictive criteria, please explain why.

Response 6: Thank you for your professional suggestion. We feel sorry for this mistake. The absolute value of log2(fold change) ≥ 1was used and we have corrected it (line 125).

Point 7: Lines 186-188: The information of the statistical approach used to data analysis is very limited. Did you check the normality of distribution of obtained data?  You used t-student test analysis which can indicate that data has a normal distribution. On the other hand, you should use a non-parametric test.  The correlation used to mRNA-lncRNA validation using qPCR should be described.

Response 7: Thank you for your professional comments. As your suggestion, we have checked the distribution of data in each group. Non-parametric test was used when the data dose not have a normal distribution. Expressional correlation analysis between lncRNA and mRNA was performed using the CORREL function by excel software. Results can be presented by scatter plot, showing the formula and r value. More details have been added to the Statistical analysis section. Details have been added in lines 190-197.

Point 8: Line 192: Did you verify why the differences in a number of reads per samples are so huge? Table S3 clearly show that young animals have average higher number of raw reads compared to adult animals (195413343 vs 147820378). I know that you normalization algorithm, but as mapping ration shows that groups differ also in % of mapping 77.8% for young and 85.2% for adults. Please explain.

Response 8: Thank you for your professional comment. To ensure the quality of tissue sample, RNA sequencing of young (6 months old) and adult (30 months old) animals were performed shortly after the animals were slaughtered (at different times). In other words, young and adult animals were separately sequenced. But their sequencing data were analyzed at the same time. We think this should be the main reason why relative large differences in the number of raw reads and mapping ratio between young and adult animals were obtained. Besides, some other factors can affect the raw reads number, such as RNA quality, cDNA concentration, and operational error. These are the possible influences that we can think of.

Reviewer 2 Report

The manuscript 'High-throughput RNA sequencing reveals NDUFC2 AS lncRNA promotes adipogenic differentiation in Chinese Buffalo (Bubalus bubalis)' describes investigations about lncRNAs and mRNAs profiles using RNA-Seq, in relationship with adipogenesis in Chinese Buffalo detailed in a first part. In a second part, the study focuses on the function of one example, NDUFC2-AS, identified in this paper in relation with adipogenic differentiation.

There are a few conclusions that the authors draw (particularly around NDUFC2-AS) that are weaker, however, and these merit further treatment in the discussion.

Introduction:

L33: I think one verb is missing in this sentence. Also, I recommend adding the information that lncRNAs are defined as transcripts of more than 200 nucleotides that are not translated into proteins.

Results:

L205: I am wondering what are "unclassified lncRNAs". Not intergenic and not intronic... So, where are they?

L224: how did you choose these ten DE mRNAs and 14 DE lncRNAs? (randomly?)

L293: Please complete "the full-length of"?

L301: Do you have any value (and p-value) to complete your sentence/legend? Also, on FIg 5b. Can you add the D0 picture of your fluorescent for oil red O? Showing evolution between D0 and D6 will be very interesting, and helpful for readers to see the difference

Discussion

I think part 4.2 and 4.3 could be merged as one part.

L335-340: You wrote: "175 lncRNAs had been identified to be specifically regulated during adipogenesis"- Are, your 124 lncRNAs identified in this study, part of the 175? did you find some common lncRNAs previously identified in other species?

L337: can you give some examples of effects to complete the sentence?

L343: Can you add references to reinforce your statement about low conservation? is there any other example of non-coding RNAs involve in the regulation of THRSPgene?

Fig 4e. You show that NDUFC2-ASis highly expressed at 10d, but Fig 5d, overexpression of it using your construction leads to weaker expression after 4d. Can you discuss about this? what is the impact on the other genes’ expression levels? Did you observe any correlation between NDUFC2-ASand THRSP expressions during the differentiation (i.e:low level of NDUFC2-AS = low level of THRSP at 2d, 4d, etc..?.). To complete this question, you did not discuss about the fact that lncRNA sequences can contain enhancer element within their sequences. By this way, the high expression level of THRSPobserved in your experiment could simply result of an enhancer element inserts in your construction, and not NDUFC2-AS. If you can discuss about this hypothesis.

Minor changes

Genes/lncRNAs names sometimes in italic, sometimes not ... please fix it.

The editor will decide but I also recommend writing NDUFC2 lncRNA like NDUFC2-AS.

L335: "...different type of adipocytes..." is more scientific.

L355-356: double "Fat deposition" written in the same sentence. I recommend " These results indicate that the candidate lncRNAs may regulate the expression of genes with key roles in fat deposition".

Fig 1. mRNA and lncRNA expressions

Fig 2: Police in Y-axis legend is different between/within each graph. In legend, please, fix "six up-regulated".

Fig 3. At the end of the legend, put the bracket please: (n=50).

Fig 4 and 5: add in the legend the *** p-value definition and named the statistic test used.

Fig 5. As previously suggested, add pictures for D0 on fig 5b. On Fig 5e, you have twice 0d in your graph. Also, as just a double-check, can you check you y-axis on fig 5f? I have never seen such scale (starting at 0.1) with expression level reaching 1000000.

Author Response

Response to Reviewer 2 Comments

The manuscript 'High-throughput RNA sequencing reveals NDUFC2-AS lncRNA promotes adipogenic differentiation in Chinese Buffalo (Bubalus bubalis)' describes investigations about lncRNAs and mRNAs profiles using RNA-Seq, in relationship with adipogenesis in Chinese Buffalo detailed in a first part. In a second part, the study focuses on the function of one example, NDUFC2-AS, identified in this paper in relation with adipogenic differentiation.

There are a few conclusions that the authors draw (particularly around NDUFC2-AS) that are weaker, however, and these merit further treatment in the discussion.

Introduction:

Point 1: L33: I think one verb is missing in this sentence. Also, I recommend adding the information that lncRNAs are defined as transcripts of more than 200 nucleotides that are not translated into proteins.

Response 1: Thank you for your kindly suggestion. We have modified the sentences in lines 32-35.

Results:

Point 2: L205: I am wondering what are "unclassified lncRNAs". Not intergenic and not intronic... So, where are they?

Response 2: Thank you for your careful review. lncRNAs can be divided into five biotypes in relation to their proximity to protein-coding genes: sense (in the sense stand and overlap with exon), antisense (in the antisense stand and overlap with exon), bidirectional (in the antisense stand and within 1 KB region of the transcription start site), intronic (in the sense stand and within the intron region), and intergenic (in the sense stand and within the intergenic region) (Sun et al. 2014). In this study, lncRNA was classified based on the order: intronic lncRNA > sense lncRNA > antisense lncRNA > bidirectional lncRNA > intergenic lncRNA. Others were unclassified lncRNAs, such as lncRNA in the antisense stand and in the downstream.

Sun T, Ye H, Wu C L, et al. Emerging players in prostate cancer: long non-coding RNAs. Am J Clin Exp Urol, 2014, 2(4):294-299.

Point 3: L224: how did you choose these ten DE mRNAs and 14 DE lncRNAs? (randomly?)

Response 3: Thank you for your careful review. Ten DE mRNAs associated with lipid metabolism and 14 DE lncRNAs were randomly selected for validation. More details were added in lines 233-234.

Point 4: L293: Please complete "the full-length of"?

Response 4: Thank you for your kindly review. We feel sorry for this mistake. Information has been completed in line 310.

Point 5: L301: Do you have any value (and p-value) to complete your sentence/legend? Also, on Fig 5b. Can you add the D0 picture of your fluorescent for oil red O? Showing evolution between D0 and D6 will be very interesting, and helpful for readers to see the difference

Response 5: Thank you for your professional suggestion. Data are presented as mean ± SD (n = 3, *p < 0.05, **p < 0.01) (lines 304 and 332). “(p < 0.05)” has been added in line 319. Oil red O staining in day 0 has been further presented in figure 5b.

Discussion

Point 6: I think part 4.2 and 4.3 could be merged as one part.

Response 6: Thank you for your professional suggestion. Part 4.2 and 4.3 have been merged as one part.

Point 7: L335-340: You wrote: "175 lncRNAs had been identified to be specifically regulated during adipogenesis"- Are, your 124 lncRNAs identified in this study, part of the 175? did you find some common lncRNAs previously identified in other species?

Response 7: Thank you for professional and careful review. lncRNAs always origin from non-coding regions and it is known that lncRNAs are with low conservation between species. Therefore, we had not compared the sequences of our lncRNAs to others. As your suggestion, we tried to blast our 124 lncRNAs to other published lncRNAs via several lncRNAs database, but it didn’t work. Then, ten of the 124 lncRNAs were randomly selected and were blasted in NCBI. We found that most lncRNAs show the highest similarity to Bos sequence (percentage identity > 90%), followed by Ovis sequence. Noncoding regions origin lncRNAs show very low similarity to sequences in other species, and coding regions origin lncRNAs show relative high similarity to sequences in other species. Only 4 of the 124 lncRNAs (Antisense lncRNA) overlap with coding regions, namely TCONS_00333050, TCONS_00307295, TCONS_00475646, TCONS_00539092 (NDUFC2-AS lncRNA). The following table shows the similarity of our 4 Antisense lncRNAs between bovine and other animals.

lncRNA

lncRNA_class

Protein_coding_within_3kb

length (bp)

Qurey coverage

Bovine

Ovis aries

Mus musculus

TCONS_00333050

Antisense lncRNA

CYTIP

2648

100%

81%

15%

TCONS_00307295

Antisense lncRNA

APPRIS P1

1651

95%

73%

4%

TCONS_00475646

Antisense lncRNA

AZGP1

2235

100%

95%

7%

TCONS_00539092

Antisense lncRNA

NDUFC2,THRSP

2493

99%

99%

13%

Point 8: L337: can you give some examples of effects to complete the sentence?

Response 8: Thank you for your kindly suggestion. We have modified the sentence in lines 355-356.

Point 9: L343: Can you add references to reinforce your statement about low conservation? is there any other example of non-coding RNAs involve in the regulation of THRSP gene?

Response 9: Thanks for your professional comment. Two references have been added in line 363. To the best of our knowledge, no other lncRNA or miRNA involved in the regulation of THRSP has been revealed.

Point 10: Fig 4e. You show that NDUFC2-AS is highly expressed at 10d, but Fig 5d, overexpression of it using your construction leads to weaker expression after 4d. Can you discuss about this? what is the impact on the other genes’ expression levels? Did you observe any correlation between NDUFC2-ASand THRSP expressions during the differentiation (i.e:low level of NDUFC2-AS = low level of THRSP at 2d, 4d, etc..?.). To complete this question, you did not discuss about the fact that lncRNA sequences can contain enhancer element within their sequences. By this way, the high expression level of THRSP observed in your experiment could simply result of an enhancer element inserts in your construction, and not NDUFC2-AS. If you can discuss about this hypothesis.

Response 10: Thank you very much for your professional comment and suggestion. Primary adipocytes are very sensitive to culture condition. Treating with adenovirus made cells grew poorly and promoted cells apoptosis. Thus, the highest expression level was found in day 10 of differentiation in normal cultured cells (Figure 4e) while that was found in day 2 of differentiation in Ad-GFP/NDUFC2-AS lncRNA cells (Figure 5d). This has been further discussed in lines 396-401. Relatively high expressional correlation between NDUFC2-AS lncRNA and THRSP was detected during differentiation of buffalo adipocytes (Figure 4f). Details have been added in Figure 4f and lines 293-294 and 304-307. We further discuss the possible regulatory mechanisms of NDUFC2- AS lncRNA during lipid accumulation in lines 410-419. We feel sorry that we do not understand the sentence very well “By this way, the high expression level of THRSP observed in your experiment could simply result of an enhancer element inserts in your construction, and not NDUFC2-AS”. Do you mean that we inserted an enhancer element in our construction? Only the full length of NDUFC2-AS lncRNA was inserted into our construction and no other element. Construction had been confirmed by sequencing.

Minor changes

Point 11: Genes/lncRNAs names sometimes in italic, sometimes not ... please fix it.

Response 11: Thank you for your careful review. We feel sorry for such a mistake. Genes/lncRNAs symbols all over the manuscript were checked and written in italic.

Point 12: The editor will decide but I also recommend writing NDUFC2 lncRNA like NDUFC2-AS.

Response 12: Thank you for your kindly suggestion. “NDUFC2-AS lncRNA” has been written as “NDUFC2-AS lncRNA” all over the manuscript.

Point 13: L335: "...different type of adipocytes..." is more scientific.

Response 13: Thank you for your professional suggestion. This has been modified in line 353.

Point 14: L355-356: double "Fat deposition" written in the same sentence. I recommend " These results indicate that the candidate lncRNAs may regulate the expression of genes with key roles in fat deposition".

Response 14: Thank you for your careful review and professional suggestion. The sentence has been modified in lines 373-374.

Point 15: Fig 1. mRNA and lncRNA expressions

Response 15: Thank you for careful review. This mistake has been fixed in line 226.

Point 16: Fig 2: Police in Y-axis legend is different between/within each graph. In legend, please, fix "six up-regulated".

Response 16: Thank you for your careful review. We feel sorry for such a mistake. “six up-regulated” has been written in line 241. We don’t understand very well for the comment “Police in Y-axis legend is different between/within each graph.” Three Y-axis title are all “Relative expression”. Expression levels of both lncRNA and mRNA are normalized to those of GAPDH.

Point 17: Fig 3. At the end of the legend, put the bracket please: (n=50).

Response 17: Thank you for your careful review. It has been modified in line 282.

Point 18: Fig 4 and 5: add in the legend the ** p-value definition and named the statistic test used.

Response 18: Thank you for your professional review. Details have been added in lines 303-304 and 331-332.

Point 19: Fig 5. As previously suggested, add pictures for D0 on fig 5b. On Fig 5e, you have twice 0d in your graph. Also, as just a double-check, can you check you y-axis on fig 5f? I have never seen such scale (starting at 0.1) with expression level reaching 1000000.

Response 19: Thank you for your professional review and kindly suggestion. We feel sorry for these mistakes. All the mistakes have been corrected in figure 5.

Round 2

Reviewer 1 Report

The manuscript has been improved according to reviewers comments and suggestions and can be published in Genes journal. 

This manuscript is a resubmission of an earlier submission. The following is a list of the peer review reports and author responses from that submission.

Round 1

Reviewer 1 Report

Comments on the manuscript Long Noncoding RNAs in Buffalo (Bubalus bubalis)  Adipose Tissues Involved in Fat Deposition

This study deals with fat deposition in a scarcely studied species, such as the water buffal. Therefor, it is interesting to read, but also it can apport interesting datato that could have a great impact on the selection of buffaloes in order to improve meat with better quality.

However, there are some problems, from my point of view, which made the study weak in general.

The principal problem is the experimental design. Only six animals were involved and they were slaughtered at two different ages. The authors assume that is enough to obtain two phenotypes divergent on adipogenesis and fat deposition. I doubt that, since differences could be due to several issues, such as age, the growth stage of the individual, metabolism of the individuals, etc.; making this analysis weak. I don’t see the point to analyze two different age stages to compare adipose deposition in spite of a case-control study in which adipogenesis could be assessed properly.

But also, from my point of view, the mat and met part is messy and not clear.

First of all, most of them assume is based on in silico analysis using cattle genomes. But also, part of the genes employed was selected arbitrarily from a bigger pool of overexpressed genes on phenotypes with no objective differentiation.

Then, the author's cultured buffalo adipocytes without a clear end. They stated that they were digested and collected for further culture. But then I did not saw that part. 

Then they tried to validate the lncRNA in cattle adipocytes. Again, I think that the species differences could be a problem. That part of the experimental design should be clarified or repeated in buffaloes tissue.

Finally, replicates description is missing in most of the experiments.

I think that the authors should clarify how the experiments were made in a clear and concise way and present the results in a more coherent way before this manuscript could be considered for publication

Please also note that I don’t make any comment on discussion since I think that the issues mentioned before should be addressed first.

Additionally, the fluidity and English quality of the manuscript is also regular. Finally, as an additional comment, I wonder if the use of UMD3.1 is not biasing the results. Despite that, that cattle genome was used before in this species I'm not sure that its use is neutral.

Overall, my opinion is that the manuscript should be considerably improved previous to be considered for publication.

Author Response

Response to Reviewer 1 Comments 

Comments on the manuscript Long Noncoding RNAs in Buffalo (Bubalus bubalis) Adipose Tissues Involved in Fat Deposition

This study deals with fat deposition in a scarcely studied species, such as the water buffalo. Therefore, it is interesting to read, but also it can apport interesting data to that could have a great impact on the selection of buffaloes in order to improve meat with better quality.

However, there are some problems, from my point of view, which made the study weak in general.

Thank you for your careful review and professional comments. We are very grateful for the professional comments to our manuscript. Careful modification has been made according to the comments, which is marked with blue color. The language of the manuscript has been further edited by professional native language editors in MOGOEDIT, changes have been marked to the text with track changes. Point by point response to reviewers is as follow.

Point 1: The principal problem is the experimental design. Only six animals were involved and they were slaughtered at two different ages. The authors assume that is enough to obtain two phenotypes divergent on adipogenesis and fat deposition. I doubt that, since differences could be due to several issues, such as age, the growth stage of the individual, metabolism of the individuals, etc.; making this analysis weak. I don’t see the point to analyze two different age stages to compare adipose deposition in spite of a case-control study in which adipogenesis could be assessed properly.

Response 1: Thanks very much for your professional comments. Generally, a higher number of sample repeat results in a more reliable data, and at least, three biological repeats are necessary. In consideration for the cost, only three repeats for each group were used in this study. Fat deposition can be influenced by heredity (breed or individual), age (development stage), and feeding and management levels (environment). Fat deposition level is different at different development stages. In general, fat is easier to deposit in adult animal than that in young animal. And, this should be regulated by a series of transcripts. For individual’s factor, adipose tissue should be sampled in vivo. However, our experiment condition is limit. Despite these weaknesses, two lncRNAs affecting lipid accumulation were obtained in our study. Similar experimental design can be found in animals (Butchart et al. 2016; Zhan et al. 2016).

Butchart LC, Fox A, Shavlakadze T, et al. The long and short of non-coding RNAs during post-natal growth and differentiation of skeletal muscles: Focus on lncRNA and miRNAs[J]. Differentiation, 2016:S0301468115300608.

Zhan S, Dong D, Zhao W , et al. Genome-wide identification and characterization of long non-coding RNAs in developmental skeletal muscle of fetal goat[J]. BMC Genomics, 2016, 17(1):666.

But also, from my point of view, the mat and met part is messy and not clear.

Point 2: First of all, most of them assume is based on in silico analysis using cattle genomes. But also, part of the genes employed was selected arbitrarily from a bigger pool of overexpressed genes on phenotypes with no objective differentiation.

Response 2: Thank you for your professional comments. Genomes between buffalo and cattle are with relatively high similarity and homology. Genome of buffalo is not available and thus, cattle genome is used as reference, which is used in other studies as well. Co-expression analysis was performed between differentially expressed mRNAs and lncRNAs (2.7 Co-expression analysis), not “selected arbitrarily”.

Point 3: Then, the author's cultured buffalo adipocytes without a clear end. They stated that they were digested and collected for further culture. But then I did not saw that part.

Response 3: Thank you for your professional and careful review. The cultured buffalo adipocytes (2.8 Cell culture) were used for the following “2.9 Adenovirus transfection” and “2.10 Adipogenic differentiation, Oil Red O staining, and quantification”. We feel sorry that we did not make it clear in these parts. In the new version, we emphasize the “primary buffalo adipocytes” in lines 159, 163 and 167.

Point 4: Then they tried to validate the lncRNA in cattle adipocytes. Again, I think that the species differences could be a problem. That part of the experimental design should be clarified or repeated in buffaloes tissue.

Response 4: Thank you for your professional and careful review. Functional validation of lncRNAs was performed by using the primary buffalo adipocytes and we further emphasize the “primary buffalo adipocytes” in lines 159, 163 and 167.

Point 5: Finally, replicates description is missing in most of the experiments.

Response 5: Thank you for your professional and careful review. All the experiments of functional validation and qRT-PCR were performed for three times. We have stated this information in the new version in lines 91-92, 160-161, 174-175, and 183-184.

Point 6: I think that the authors should clarify how the experiments were made in a clear and concise way and present the results in a more coherent way before this manuscript could be considered for publication

Response 6: Thank you for your professional suggestion. Careful modification has been made for the “Materials and Methods” and “Results” parts.

Point 7: Please also note that I don’t make any comment on discussion since I think that the issues mentioned before should be addressed first.

Response 7: We greatly appreciate the professional and valuable comments to our manuscript and careful modifications have been made in the new version.

Point 8: Additionally, the fluidity and English quality of the manuscript is also regular. Finally, as an additional comment, I wonder if the use of UMD3.1 is not biasing the results. Despite that, that cattle genome was used before in this species I'm not sure that its use is neutral.

Response 8: Thank you for your professional comments. The language has been further edited by professional native language editors in MOGOEDIT. We have realized the weakness of this manuscript. However, buffalo’s genomic information is not available and we can only use the cattle genome as reference. 5’RACE and 3’RACE were performed to confirm the existence of the two lncRNAs before adenovirus packaging in our manuscript (data were not showed). Importantly, two lncRNAs promoted lipid accumulation in primary buffalo adipocytes were obtained. All these results indicate the feasibility of our method.

Reviewer 2 Report

Comments and Suggestions for Authors

Introduction:

Line 32-33:  In China, the buffalo account 32 for 20% of the total number of “cattle”. Cattle and buffalo are different. Please replace “cattle” with bovines or Bovidae.

Line 36: what is buffalo meat called. Put in parenthesis as you have put for cattle meat (beef).

Line 39-40: The exploration 39 of significant internal effectors that are involved in fat deposition will provide biomarkers for future 40 buffalo breeding. Let us be realistic about selecting buffalo for harboring a given lncRNAs you have found. Your studies value is that you found two candidate lncRNAs for fat deposition, hence future is to make a way for their enhanced expression in buffaloes of china. Please replace this whole sentence accordingly.

Line 60: via should be italicized.

Material and methods:

Results:

1.      Fig 1a: Cluster graph of all mRNAs and lncRNAs based on correlation analysis. The correlation of mRNA and lncRNA expression between both groups indicated the reliability of RNA sequencing 215 performance. Please make all the axis legends easy to read. Also, do you consider the 0.78 is a strong correlation.

2.       Fig. 1b: Classification of 9,494 lncRNAs is not readable. Please replace this figure. Make all the classes readable.

3.      Fig. 1c: Hierarchical clustering 216 of 2,008 significantly differentially expressed (SDE) mRNAs from young and adult groups. I would have like to see the repeat of this experiment, to see how the repeat 1 and repeat for each young and adult SDE are matching. If possible please do that.

4.       Fig. 1d: Hierarchical clustering of 124 SDE lncRNAs from young and adult groups. Same above comment for this figure.

5.      Fig 2a, b, c, d: Please replace and provide with high resolution and axis should be readable.

6.      Fig 3a, 4a: Please decrease the gain of GFP channel. I know adenovirus did good transfection of your construct.

Over and above: I would have liked to see Result 3.6: Co-transfection of TCONS_00539210 and TCONS_00539092 on lipid accumulation in bovine adipocytes

Overall recommendation: Increasing buffalo meat palatability by improving fat deposition in adipocytes is needed and I appreciate the research groups work. However, I would like to see mechanism in a picture form how lncRNAs are responsible for improving fat deposition in adipocytes. Group should also conduct live animal experiments to inject their two lncRNA constructs and to see the increase in fat deposition.

Accept after revisions of my review points.

Author Response

Response to Reviewer 2 Comments

Comments and Suggestions for Authors

Thank you for your careful review and professional comments. We are very grateful for the professional comments to our manuscript. Careful modification has been made according to the comments, which is marked with blue color. The language of the manuscript has been further edited by professional native language editors in MOGOEDIT, changes have been marked to the text with track changes. Point by point response to reviewers is as follow.

Introduction:

Point 1: Line 32-33: In China, the buffalo account for 20% of the total number of “cattle”. Cattle and buffalo are different. Please replace “cattle” with bovines or Bovidae.

Response 1: Thank you for your professional comments. We have modified the sentence.

Point 2: Line 36: what is buffalo meat called? Put in parenthesis as you have put for cattle meat (beef).

Response 2: Thank you for your careful review. We failed to fine a native English name for “buffalo meat”.

Point 3: Line 39-40: The exploration of significant internal effectors that are involved in fat deposition will provide biomarkers for future buffalo breeding. Let us be realistic about selecting buffalo for harboring a given lncRNAs you have found. Your studies value is that you found two candidate lncRNAs for fat deposition, hence future is to make a way for their enhanced expression in buffaloes of china. Please replace this whole sentence accordingly.

Response 3: Thank you for your professional comments. We have realized that the sentence overstating the value of this study. Thus, we delete this sentence.

Point 4: Line 60: via should be italicized.

Response 4: Thank you for your professional suggestion. “via” has been italicized in the manuscript.

Material and methods:

Results:

Point 5: Fig 1a: Cluster graph of all mRNAs and lncRNAs based on correlation analysis. The correlation of mRNA and lncRNA expression between both groups indicated the reliability of RNA sequencing performance. Please make all the axis legends easy to read. Also, do you consider the 0.78 is a strong correlation?

Response 5: Thank you for your professional comments. In this study, two groups were set, namely young group (n = 3) and adult group (n = 3). Correlation values between two of the three samples within a group were higher than those out of the group, which indicated good repeatability of the sample within group.

Point 6: Fig. 1b: Classification of 9,494 lncRNAs is not readable. Please replace this figure. Make all the classes readable.

Response 6: Thank you for your professional suggestion. Figure 1b has been replaced.

Point 7: Fig. 1c: Hierarchical clustering of 2,008 differentially expressed (DE) mRNAs from young and adult groups. I would have like to see the repeat of this experiment, to see how the repeat 1 and repeat for each young and adult DE are matching. If possible please do that.

Response 7: Thank you for your professional suggestion. Figure 1c has been constructed again and the result meet with previous result roughly.

Point 8: Fig. 1d: Hierarchical clustering of 124 DE lncRNAs from young and adult groups. Same above comment for this figure.

Response 8: Thank you for your professional suggestion. Figure 1d has been constructed again and the result meet with previous result roughly.

Point 9: Fig 2a, b, c, d: Please replace and provide with high resolution and axis should be readable.

Response 9: Thank you for your professional suggestion. All the figures are 500 dpi and should be inserted into the text according to the requirements of Journal “Genes”. The JPEG or PDF file of each figure with high resolution will be submitted at last. The axis indicates the relative expression of lncRNA (to GAPDH) and THRSP (to GAPDH). Details of qRT-PCR and analysis are provided in the part “2.11” and “3.4”.

Point 10: Fig 3a, 4a: Please decrease the gain of GFP channel. I know adenovirus did good transfection of your construct.

Response 10: Thank you for your professional comments. Repeating this experiment will take about a month and our time is very limit (10 days). We will decrease the gain of GFP channel in our further study.

Point 11: Over and above: I would have liked to see Result 3.6: Co-transfection of TCONS_00539210 and TCONS_00539092 on lipid accumulation in bovine adipocytes

Response 11: Thank you for your professional review. This is a good idea. This experiment will take about 2-3 months (isolation and culture of bovine primary adipocytes, co-transfection, Oil Red O staining, qRT-PCR, with at least two repeat experiments). However, time is limit (10 days). But anyhow we will try to co-transfect the two lncRNAs to estimate their effect on lipid accumulation in bovine adipocytes in our further study.

Point 12: Overall recommendation: Increasing buffalo meat palatability by improving fat deposition in adipocytes is needed and I appreciate the research groups work. However, I would like to see mechanism in a picture form how lncRNAs are responsible for improving fat deposition in adipocytes. Group should also conduct live animal experiments to inject their two lncRNA constructs and to see the increase in fat deposition.

Response 12: Thank you for your professional review and suggestion. Presently, specific regulatory mechanisms of lncRNAs TCONS_00539210 and TCONS_00539092 on lipid accumulation in adipocytes have not been studied and this is in our plan, including the live animal experiments.

Round 2

Reviewer 1 Report

Comments on the manuscript Long Noncoding RNAs in Buffalo (Bubalus bubalis)  Adipose Tissues Involved in Fat Deposition R1

First of all, it is very unlikeble that the authors could assess a major review of the manuscript on only 10 days. It is possible, but is unlikeable.

In this case, its was possible since the authors did not correct most of my comments and did not modify almost nothing of the manuscript. They just justify the absence of replicates and the weak experimental design apologizing and citing a couple of similar papers. This is not an aenough explanation for me. So my doubts regarding to the experimental desing and the lack of replicates and the absence of a clear phenotype remains.

But also, there is a big additional problem that I was not aware in the previous review. Buffalo genome (UOA_WB_1) was publicly released in January of this year (https://www.nature.com/articles/s41467-018-08260-0). Therefore, the claims of the authors regarding the absence of buffalo genomic information are unsustained. I am sorry that I did not check this point before in my previous review letter. But also I believe that the authors should be aware if the genomic information of their species of interest is available or not.

Finally, I don’t understand the use of cattle or buffalo adipocite lines in the validation. Abstract remains saying that cattle adipocites were used. Answering letter say it is not.

Therefore, the manuscript should be susbtanstally modified before be able to be considered.

My opinion is that it should be rejected in the present form, but I also believe that a new manuscript using the new information available could be interesting.